# A Review of Laboratory Requirements to Culture Lichen Mycobiont Species

**DOI:** 10.3390/jof10090621

**Published:** 2024-08-30

**Authors:** Dania Rosabal, Raquel Pino-Bodas

**Affiliations:** 1Medical Science Faculty, Saint Joseph University, Roseau ZC 00109-8000, Dominica; 2Biologia, Geologia, Fisica and Quimica Department, Universidad Rey Juan Carlos, 28008 Madrid, Spain; raquel.pino@urjc.es; 3Instituto de Investigación en Cambio Global de la Universidad Rey Juan Carlos (IICG-URJC), 28008 Madrid, Spain

**Keywords:** mycobiont culture, culture media, cultivation requirements

## Abstract

Lichens are symbiotic associations between fungi (the mycobiont) and algae or cyanobacteria (the photobionts). They synthesize a large number of secondary metabolites, many of which are potential sources of novel molecules with pharmacological and industrial applications. The advancement of in vitro culture methods of lichen-forming fungi would allow the comprehensive application of these compounds at large scales, enable improvements in the synthesis, facilitate understanding of the role of the partners in the synthesis of these compounds and increase our knowledge about the genes associated with secondary metabolites production. The aim of this work is to summarize the nutritional and physicochemical requirements that have been used to date to culture different lichen-forming fungi species. In total, the requirements for the cultivation of 110 species are presented. This review can provide a starting point for future experiments and help advance the methods of culturing lichenized fungi. The type of diaspore selected to isolate the mycobiont, the composition of the isolation and culture media and the corresponding physicochemical parameters are essential in designing an efficient lichen culture system, allowing the achievement of a suitable growth of lichen-forming fungi and the subsequent production of secondary metabolites.

## 1. Introduction

Lichens are symbiotic associations between nutritionally specialized and ecologically obligate biotrophic fungi (the mycobionts), which take up fixed carbon from algae or cyanobacteria (the photobionts) [1]. Lichens are one of the most successful examples of symbiosis in nature; approximately 17% of all fungi are lichen-forming fungi [2], occupying 8% of the Earth’s surface and capable of living in the most extreme habitats, such as hot and cold deserts—regions where most organisms find it difficult to survive. Additionally, lichens are rich in secondary metabolites, more than 700 of which have been described so far [3,4]. The biological role of many of these compounds is not yet well understood, but they should contribute significantly to the fitness of lichens given that secondary metabolites account for 0.1% to 10% of the dry weight of the thallus [5]. Some of them provide photoprotection for the algal layer or protection against microorganisms or grazing by invertebrates, help tolerate extreme ecological conditions or contribute to the competitive capabilities of lichens [6,7,8,9,10,11,12,13,14,15,16,17,18,19,20,21,22,23,24]. On the other hand, lichen substances are potential sources of novel molecules with antibacterial, antiviral, antifungal, anticancer, antipyretic, anti-inflammatory, analgesic and enzyme inhibition activities [25,26,27,28,29,30,31,32,33,34,35,36,37].

Despite this, our knowledge of their biology is far less than that of other organisms, including non-lichenized fungi. Although numerous studies have successfully cultured both mycobionts and photobionts and carried out re-synthetic experiments, the difficulties of culturing lichens remain a serious handicap, limiting progress in gaining biological knowledge of them and their industrial and pharmacological applications. 

The advent of genome sequencing has facilitated the identification of genes involved in the biosynthesis of secondary metabolites in lichens [38]. These genes are organized into gene clusters (BGCs) that include as many as 20 genes. Genomic studies have been carried out to identify biosynthetic gene clusters (BGCs) encoding the production of atranorin, anthraquinones, grayanic, gyrophoric, lecanoric, olivetoric, usnic acids, and biruloquinone [39,40,41,42,43,44]. The exploration of biosynthetic gene content provides insights into the genomic adaptations which lead to evolutionary success. The development of in vitro culture methods for lichen-forming fungi would facilitate the large-scale synthesis of secondary metabolites, enhance our understanding of the role of partners in their synthesis, and increase our knowledge of the genes connected to secondary metabolite production, thereby improving heterologous expression systems.

Several culture experiments have been performed to determine the optimal culture conditions for many lichen species. Physicochemical in vitro conditions such as temperature, photoperiod, pH, UV radiation, humidity and nutrient composition have been proven to influence secondary metabolite production in lichens [45,46,47]. The ability of lichen-forming fungi species of the same genus to flourish in a range of ecological environments represents a significant challenge for the standardization of methods and techniques for lichen culturing at the genus level. Therefore, experimental lichenological studies focus on the culture conditions and optimization of techniques in target species more than in lichen genera. 

The aim of this study is to summarize the nutritional and physicochemical requirements that have been used to date to culture different lichen-forming fungi species. This paper points out aspects that must invariably be taken into account for the appropriate design of experiments in the cultivation of mycobionts. In addition, it allows a global perspective of the techniques and conditions that have been used for the cultivation of different lichen species. Additionally, this review can provide a starting point for future experiments and help advance the methods of culturing lichen-forming fungi.

## 2. Materials and Methods

A systematic revision of articles was carried out using digital data sources of wide scope and international prestige (Google scholar, Scielo, DOAJ, Latiendex and SCOPUS). 

Reference searching was conducted using the following keywords: “lichen culture”, “lichen metabolites production”, and “culture media for lichen”. The selected scientific articles met the following inclusion criteria: the description of the mycobiont isolation methods, the isolation and culture media and physicochemical conditions. Additionally, the secondary metabolites production from cultured species was documented, but it did not constitute an inclusion or exclusion criterion.

A total of 30 relevant studies were identified for inclusion in the review, having met the pre-stablished criteria. This paper recorded the culture conditions for 110 lichen-forming fungi. 

## 3. Overview of Mycobiont Cultures

This review focused on cultures of mycobionts belonging to Ascomycota. However, Oberwinkler [48] summarizes all the studies that have performed mycobiont cultures of basidiolichens. Both the isolation and successful cultivation of *Omphalina* and *Botrydina* species have been achieved. 

In lichenology, experimental development has focused on certain fundamental aspects, such as lichen re-synthesis and the obtaining of “tissue cultures” of lichens. Following the discovery of the dual nature of lichens by Schwendener in 1867, efforts were made to conduct re-synthesis experiments, which entailed the cultivation of both symbionts. The initial successes were achieved by several researchers, who successfully re-synthesised cultured fragments of lichen thalli and described the growth of the photobiont when isolated from the mycobiont. However, most of the attempts to resynthesize lichen from separate components in the late 19th century did not develop beyond that preliminary stage. 

The major development of lichen cultivation methods peaked in the 1980s and 1990s, led by Yamamoto et al., Yoshimura et al. and Ahmadjian, among others [49,50,51]. Subsequently, this research was pushed into the background, mainly because of its difficulty. Nowadays, with the search for new metabolites of interest in biomedicine and the development of genomics, there has been a new surge in attempts to cultivate lichen symbionts. 

This review presents a summary of conditions under which 110 species from 11 orders have been successfully cultivated (Table A1). The majority of species belonged to the Lecanorales (52) order, followed by Ostropales (16). The *Parmeliaceae* was the family with the largest number of species studied (22). For the majority of species, the data were derived from a single study. However, for some species such as *Xanthoria parietina* (L.) Th. Fr. data from four different studies exist. The genera with the greatest number of studied species were *Graphis* (10), *Ramalina* (9), *Cladonia* (9) and *Lecanora* (8) (Table A1). In this work, most of the cultivated species were chlorolichens, while the number of cyanolichen species was relatively low (about 6%). 

## 4. Mycobiont Isolation

The methods described by Ahmadjian [49] and Yoshimura et al. [50] for ascospore-derived cultures have been the most utilized by researchers to isolate the mycobionts (Figure 1A). Generally, in ascospore-derived cultures, the thalli collected were surveyed under a dissecting microscope, and then thalli with mature ascomata were selected. The rehydrated ascomata were thoroughly cleaned with water under a dissecting microscope. The surfactants were removed by washing the ascomata in double distilled water. Excess water was removed with the help of filter paper, and clean ascomata were attached to the sterile petri plate lids, using petroleum jelly. Petri plates containing solidified media were then inverted over the lids, and ascospores were allowed to discharge onto the agar medium [49,50]. Mycobiont cultures from isolated ascospores exhibit low rates of contamination. However, small mycelia are produced, so long incubation periods are required.

An alternative methodology was described by Yamamoto et al. [51] for the cultivation of mycobionts derived from thallus fragments that is still used by most researchers (Figure 1B). The method of thallus fragment-derived culture (“lichen tissue culture”) was originally designed to obtain the mycobiont of lichen species that do not produce ascomata but it has been also used to isolate the mycobiont from those species whose ascospores do not germinate. Following this method, a fragment of about 1 cm in length is separated. The thallus fragment is then homogenized by maceration in distilled water and filtered through a succession of nylon sieve meshes (500 μm to 150 μm). The thallus fragments thus obtained in the mesh are picked up with the sterile needle under the dissecting microscope and inoculated onto the surface of culture medium [51].

The most successful and frequent method used for mycobiont isolation was via isolation from ascospores (53%). Records of 38 species that have been isolated from both spores and thallus fragments were found. The mycobionts of *Parmotrema reticulatum* (Taylor) M. Choisy and *Usnea orientalis* Motyk have been isolated only via the method of thallus fragment-derived culture. Mycobiont isolation from vegetative propagules has been less commonly employed. *Cladonia fimbriata* (L.) Fr., *Cl. macilenta* Hoffm, *Cl. metacorallifera* Asahina and *Cl. rei* Schaer. have been isolated from soredia. *Dirinaria applanata* (Fée) D.D. Awasthi has been isolated from ascospores, tallus fragments and soredia (Table A1). In the cyanolichen species, the mycobionts have been successfully isolated from ascospores in several species, except for *Peltigera dactyla* (With.) J.R. Laundon, whose mycobiont has been isolated from soredia and thallus fragments. 

The method of mycobiont isolation from a thallus fragment has increased its success with advances in cleaner work benches and inoculation techniques. Among the described advantages of this method are the following: (1) it does not depend on reproduction structures; (2) only small amounts of lichen material are needed; (3) the growth rates are faster than for those resulting from spore cultures [52]. Zakeri et al. [53] aimed to identify the most suitable methods for the isolation of six species by using a range of methods and media combinations. These authors concluded that isolation from ascospores is the most successful one and argue that ascospore discharge is a very suitable method, easy to handle and rarely leading to contamination. 

However, the data indicate that the culture success rate of the mycobiont derived from thallus fragments is quite high, and that it could be an alternative for use with generally sterile species [54,55]. If sporulation is chosen as the isolation method, attention should be paid to the possible seasonality of ascospore release, which has been demonstrated in many species [14,56,57,58,59,60]. In any case, it is necessary to carry out experiments to compare the growth rates of the species obtained via different isolation methods in order to select the most suitable mycobiont isolation method.

Mycobiont cultures of both ascospores and thallus fragments are susceptible to contaminants of various origins, such as fungi and bacteria. These contaminants can either come from organisms associated with the lichen (in cultures derived from thallus fragments or vegetative propagules) or be stimulated by the culture media used, in which these contaminants can sometimes flourish. The authentication of symbionts through sequencing the ribosomal genes (such as ITS rDNA, the barcode of fungi) and comparing the resulting sequences with those from the UNITE and NCBI (or alternative databases) is required to ensure that the fungi obtained in culture are the desired mycobionts. This procedure must be performed before continuing with the mycobiont culture after successful isolation [52,61]. Molecular genetic tests to confirm the species were carried out on only 40% of the mycobiont cultures (Table A1).

## 5. Isolation Media

It is common practice to use different media for the isolation of mycobionts and for the growth of mycelia. The composition of the isolation medium is critical for diaspore germination or growth. Media as Bold Basal Medium (BBM), Malt Yeast extract (MY) and plain agar were the most used (31, 54 and 58% of the species respectively), especially for ascospores germination (Table A1). Out of the seven species whose mycobionts were isolated from soredia, five used BBM as the growth medium. For mycobionts isolated from thallus fragments, the medium most often used was LB (21% of the species). 

According to Ahmadjian [49], ascospore germination success is higher on plain agar for most of the species. Other authors have also proven that the germination rate of ascospores is higher in poor nutrient media. BBM is a nutrient-poor medium, with different inorganic compounds, which is optimal for algal partner cultivation. Zakeri et al. [53] found that BBM was the optimal medium for isolation of the mycobiont in most of the species tested, with the exception of the *Cladonia* species. Species of this genus were successfully cultured on media such as Sabouraud-2%-glucose-agar, Sabouraud-4%-glucose-agar and Malt-yeast-extract- agar [52]. 

The inhibition of ascospore germination has been proven for some species in particular media: for instance, in the Gelrite medium for *Letharia columbiana* (Nutt.) J.W. Thomson. For other species—for instance, *Megalospora tuberculosa* (Fée) Sipman and *Myelochroa entotheiochroa* (Hue) Elix & Hale—the rate of germination in this medium is lower than in others. Other experiments have proven that glucose inhibits ascospore germination [62]. Germination of cyanolichen ascospores is difficult [54,63] and has only been achieved in a few species [58,59,64]. The addition of substances that can absorb phenolic compounds to culture media increased the germination of cyanolichen ascospores [58,59].

For most of the species, ascospore germination occurred within a wide range of temperatures from 3 to 25 °C, although some species showed a decrease in germination at 25 °C (*Pyxine endochrysina* Nyl.) and others at temperatures higher than 15 °C (*Graphina soozona* Zahlbr). Experiments were conducted to test the effects of different pH values on ascospore germination. The results indicated that pH 6 was the optimal value for most species, while alkaline conditions inhibited ascospore germination in some species [58].

## 6. Culture Media

Once isolation of the mycobiont from the diaspores has been achieved, the mycobiont is usually transferred to another medium for mycelial development (culture media). The nutritional condition has a direct influence on the growth of the mycobiont. The composition of many of these media has been compiled by Muggia et al. [45]. Most of these media are enriched with carbon sources such as glucose, sucrose and polyethylene glycol, and less frequently with ribitol, mannitol and sorbitol. Nitrogen sources such as amino acids and vitamins—as well as minerals—are often added to the culture medium, though they are less effective at promoting the growth of cultured mycobionts than the carbon sources [65,66]. In general, the most commonly employed media for culturing mycobionts are as follows: Malt Yeast (MY), for 60% of the species, and Lilly and Barnett (LB), for 35% of the species. In many cases the MY and LB were also enriched with glucose, sucrose, mannitol, sorbitol and other N/C sources. Those media were employed independently of the type of diaspore used to isolate the mycobiont. 

The growth rates of mycobionts are data that are usually missing from most articles, and in those that do provide them, not enough comparisons have been made to know which is the optimal growth medium for each species. The cultures of species studied in different experiments vary not only in their composition but also in other conditions, which makes it challenging to compare them (Table A1). However, a few studies compared the growth rates of mycobionts in different media. For instance, Molina et al. [61] found that *Anaptychia ciliaris* (L.) Flot. had a better growth rate in corn meal agar (CMA) in the early stages of development, similar to Lilly and Barnet medium (LBM) enriched with 3% glucose in later development stages, thereby increasing biomass production. *Dirinaria applanata* grew better on MS 4% sucrose than on MBB 4% sucrose [67]. Also, *Ramalina celastri* (Spreng.) A. Massal., *R. complanata* (Sw.) Ach. and *Xanthoria parietina* had the best performance when growing on LBM 4% glucose [68,69]. Studies that measure the growth rates of the species under a combination of isolation and culture media and physicochemical parameters provide a lot of useful information for subsequent research, in which cultivation conditions can be improved based on the results obtained by other authors. 

## 7. Physicochemical Conditions for Culture of Mycobionts

In accordance with the findings of the literature review, there are no standard physical and chemical conditions for growing mycobionts, although temperature, pH, light, and humidity have been suggested as the major driving forces for the culturing of lichens [70]. In most of the studies reviewed, temperature and photoperiod are variables frequently considered, while light intensity, pH and relative humidity are less frequently recorded (Table A1). However, not all of these variables are considered by all the authors, nor have comprehensive experiments been carried out to show which conditions are suitable for each of these variables in the different species of lichen-forming fungi. 

The mycobionts were cultivated in dark conditions in several experiments, probably to avoid the contamination of the cultures. Some researchers frequently used photoperiods of light/darkness in thallus fragment-derived cultures (14 h/10 h, 16 h/8 h). Light is a physical parameter that influences the growth of mycobiont thallus in one of two ways: namely, time of exposition or photoperiod and light intensity. The photoperiod has been included in studies of mycobiont cultures more frequently than the light intensity [60,71,72,73,74,75,76].

The most important physicochemical parameters for mycobiont cultures are probably temperature, pH, light and relative humidity [70]. However, more experiments are required for most of the relevant parameters in order to yield standard protocols for mycobiont culture.

## 8. Secondary Metabolites Production of Mycobiont Cultures

Although secondary metabolite production was not an objective of this review, they were annotated whenever these data were registered by the authors. The production of secondary metabolites in axenic culture has achieved a high level of success by means of modifications to the physicochemical conditions, such as exposure to high light intensities of UV-C, alternating desiccation periods of the medium, temperature and osmotic stress. Also, this development in secondary metabolite production has been accomplished by improving media composition, adding mannitol and others carbon sources [37,75,77,78,79,80]. In enriched culture media, some species exhibit enhanced growth and higher secondary metabolite production [81]. However, other lichen-forming fungi species grow slowly and only synthetize secondary metabolites when some nutrients become limited [70]. It is also notable that some articles lack information regarding the number of days required for the production of lichen compounds. Providing this information would facilitate the optimization of time and resources in the experimental process.

Stocker-Worgöter [52] demonstrated that, contrary to previous assumptions, mycobionts are able to produce secondary metabolites under specific culture conditions. According to her results, the production of secondary metabolites is dependent on a number of factors, including the presence of adequate nutritional resources in conjunction with appropriate ecological conditions, such as temperature, light, and humidity [75,77]. Many researchers do not provide evidence that secondary metabolites are produced by the mycobiont. This procedure must be included in the design of the experiment in all research that aims to obtain and quantify lichen metabolites.

In the present review, the secondary metabolites recorded in the mycobiont cultures are the following: norstictic acid for *Buellia* sp. and *Graphis handelii* Zahlb [81]; atranorin was found in mycobiont cultures of *Buellia* sp., *Dirinaria applanata* (Fée) D.D. Awasthi and *Lecanora niponica* H. Miyaw. [67,81]; divaricatic acid and sekikaic acid were produced in *D. applanata* cultures [67]; stearic acid, linoleic acid and oleic acid were found in cultures of *Physconia distorta* (With.) J.R. Laundon [82]; protocetraric acid, hypoprotocetraric acid, salazinic acid, consalazinic acid and usnic acid were produced in *Ramalina farinacea* (L.) Ach. cultures [72]; usnic acid was found in culture of *Usnea ghattensis* G. Awasthi [73]. The measurement of the quantities of lichen compounds obtained would facilitate the comparison of the optimal conditions for the culture and the increase in the quantities of targeted secondary metabolites.

## 9. Conclusions

The success of cultures of lichen-forming fungi depends on critical factors, including the method of isolation of the mycobiont, the isolation medium, the culture medium, and the physicochemical conditions. The mycobiont isolation method from ascospores is the most frequently chosen for lichen culturing. The correct isolation of the mycobiont must be confirmed using DNA sequences. Regarding isolation media, the BBM medium is the most used. However, MY and LB media are more frequently used as culture media for mycelium growth. The physicochemical parameters that are taken into account in most studies are temperature, pH and photoperiod. The selection of the physicochemical parameters as well as the isolation and culture media chosen are generally not justified in the reviewed literature, so we recommend exposing the selection criteria for the culture conditions. Mycobiont growth measurements must be provided by researchers, as this would allow comparison of the growth results of a species under different culture conditions. Nevertheless, research in this field remains limited, and for the majority of species no data are available. In order to utilize the secondary metabolites of lichens for a variety of applications, further experimentation is required so as to ascertain the optimal cultivation conditions for each species.

## Figures and Tables

**Figure 1 jof-10-00621-f001:**
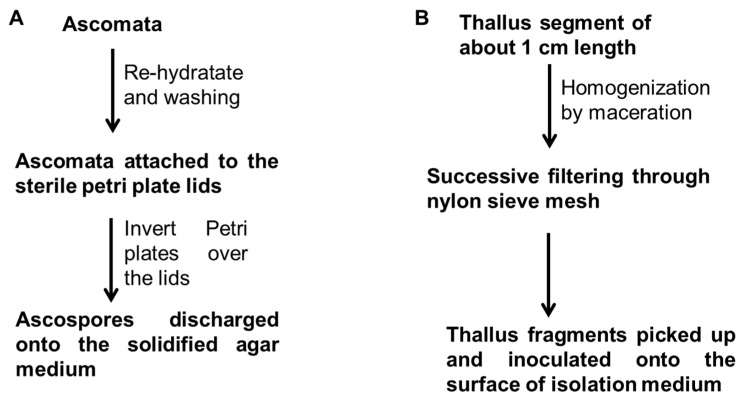
Flow chart to show the main methods of mycobiont isolation, ascospore-derived cultures (**A**) and thallus fragment-derived cultures (**B**).

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
