# Peer review of "A Review of Laboratory Requirements to Culture Lichen Mycobiont Species"

_jof, 2024, doi:10.3390/jof10090621_

Round 1

Reviewer 1 Report

In general, the manuscript is about very important topic -culturing of lichen-forming fungi, but in my mind, this paper has a lot of space for improvement. What I miss the most is visuality. Text accompanying with relevant graph for each chapter (or even for section) enhances usability of this manuscript. The manuscript points on several shortcomings in design, data performance and missing information (growth rates, parameters, etc.) in culture studies, and if you prepare a general synthesis and propose what background data to provide, the readability rises too. It will be also nice to read an analysis about the research questions what people want to know by using fungal cultures. Also, how reseraches should improve their presentation of data, And finally, I would expect a bit more synthesis / conclusions based on your research i.e. the most important conclusion from each eight chapters.

There are also a lot of typing errors, and some sentences are grammatically incorrect.

Introduction

There are a lot of information about lichen substances. Why? If one of the aims is to select techniques enhancing production of lichen substances, then it should be mentioned as an aim of the article. In the moment the space dedicated to lichen substances is enormously high. I would leave out the text starting from “Additionally” and ending “… in vitro cultures” and would concentrate in culturing of lichen-forming fungi. 

M & M

Why only screening of papers from such short period i.e. 2020-2023? Culturing of lichen-forming fungi is not smth new and has been done already more than a century. There are a couple of nice papers giving overview about techniques like that of Yoshimura et al. in book by Kranner et al. (ed). Protocols in Lichenology, 2002 or Crittenden et al. 1995 or Ahmajian’s papers. 

Most of the provided information could be additionally illustrated with graphs. In the moment, it is rather difficult to understand what is in reality new in this paper?

Overview of mycobiont cultures

Most of the cultivated fungi belong to Lecanoromycetes, but are there fungi from other classes too? Like Arthoniomycetes, Coniocybomycetes, etc. Is there difference between species having foliose, fruticose or crustose thallus?

Mycobiont isolation

Is it known how old specimens can be used for isolation? What is the optimal age i.e how quickly should be isolation done. 

Section starting from L. 149 deals with comparison of different isolation methods. But I still do not understand what are the benefits / drawbacks for one or another isolation method. In short, my understanding is that from thallus fragment it is easier to get cultures but the risk to get multiple fungi (or bacterial, etc. contamination) is very high, but from ascospores it is easier to get pure cultures, but the success rate is much lower (because of sporulation seasonality, etc.).

Section starting from l. 200. You talk about temperatures. How much the studies have taken account where the lichen comes from – high or low latitudes? The same about pH (and humidity also). Did the studies took account of original pH of lichen substrate (I mean calcareous substrate and bark of deciduous trees vs. acidic substrata and trees with bark of low pH value). 

Culture media

It will be nice to compare graphically the isolation success depending on medium with different nutrition composition. i.e. from Corn Meal to Potato dextrose to …. Is there any correlation about the composition of the medium and the substratum where the lichen grows or what photobiont(s) it uses. I guess no.

Conclusions

Conclusions are too short and for me does not carry necessary information. I would take all chapters and include point-by-point the most important message from the chapters. 

 The most valuable part of the ms is the Table, however, I would do following modifications: 1) replace the column either that of order or family with the column of the class. 2) It would be easier to follow and it will be more useful if physico-chemical parameters are divided between different columns (temperature, light, pH, etc. ), also percentage of sugar can be in separate column. 

 The style of references’ list should be unified. Check the style in the instructions of the journal. 

Author Response

August 12, 2024

Dear Reviewer 1,

We appreciate all your comments about the manuscript, which help us improve our work.

(R1) Major comments:

In general, the manuscript is about very important topic -culturing of lichen-forming fungi, but in my mind, this paper has a lot of space for improvement. What I miss the most is visuality. Text accompanying with relevant graph for each chapter (or even for section) enhances usability of this manuscript. The manuscript points on several shortcomings in design, data performance and missing information (growth rates, parameters, etc.) in culture studies, and if you prepare a general synthesis and propose what background data to provide, the readability rises too. It will be also nice to read an analysis about the research questions what people want to know by using fungal cultures. Also, how reseraches should improve their presentation of data, And finally, I would expect a bit more synthesis / conclusions based on your research i.e. the most important conclusion from each eight chapters.

There are also a lot of typing errors, and some sentences are grammatically incorrect.

(A) In general, the revision of the writing in English was carried out, the importance of the work as a unifying document of the main aspects related to the conditions for the cultivation of mycobionts was highlighted and the conclusions were improved in their content. A figure was incorporated in chapter 4 and the information on lichen metabolites was reduced.

(R1) Detail comments:

Introduction

There are a lot of information about lichen substances. Why? If one of the aims is to select techniques enhancing production of lichen substances, then it should be mentioned as an aim of the article. In the moment the space dedicated to lichen substances is enormously high. I would leave out the text starting from “Additionally” and ending “… in vitro cultures” and would concentrate in culturing of lichen-forming fungi.

(A) The information dedicated to lichen metabolites was reduced.

M & M

Why only screening of papers from such short period i.e. 2020-2023? Culturing of lichen-forming fungi is not smth new and has been done already more than a century. There are a couple of nice papers giving overview about techniques like that of Yoshimura et al. in book by Kranner et al. (ed). Protocols in Lichenology, 2002 or Crittenden et al. 1995 or Ahmajian’s papers.

(A) The work was carried out taking into account important research with those suggested by the reviewer. Perhaps the period in which this review was carried out was misunderstood. So this aspect was worded differently

(R1) Most of the provided information could be additionally illustrated with graphs. In the moment, it is rather difficult to understand what is in reality new in this paper?

(A) The novelty of this work is that it puts different investigations into perspective and allows us to point out aspects of the design that the researchers have not taken into account. Some species have been successfully cultivated, however, other researchers have not taken these experiences into account when cultivating the same species. The successful cultivation of the mycobiont is not enough; a measure of this success must be had so that other researchers can compare their results. This work suggests that this growth measure be invariably included in research. In our review we realized that very few do it

Overview of mycobiont cultures

(R1) Most of the cultivated fungi belong to Lecanoromycetes, but are there fungi from other classes too? Like Arthoniomycetes, Coniocybomycetes, etc. Is there difference between species having foliose, fruticose or crustose thallus?

(A) In our review we found no evidence of differences in culture conditions depending on the class or biotype. The selection criteria for these species by the researchers do not indicate that the class or biotype has been taken into account for their experimental designs, so it is not possible to reach conclusions in this regard.

However, this concern could become a research question for which an appropriate experimental design would be necessary to reach conclusions.

Mycobiont isolation

(R1)Is it known how old specimens can be used for isolation? What is the optimal age i.e how quickly should be isolation done.

(A) Isolation should be done as soon as possible after collection, however in some works this data is missing, in others a lot of time has passed since collection and the specimens have been preserved in different ways.

(R1) Section starting from L. 149 deals with comparison of different isolation methods. But I still do not understand what are the benefits / drawbacks for one or another isolation method. In short, my understanding is that from thallus fragment it is easier to get cultures but the risk to get multiple fungi (or bacterial, etc. contamination) is very high, but from ascospores it is easier to get pure cultures, but the success rate is much lower (because of sporulation seasonality, etc.).

(A) The success of one isolation method over another depends on several factors. If the species does not sporulate, the cultivation method from thallus fragments will be the only option, however more comparative studies of which method is better for a species under certain conditions should be carried out.

(R1) Section starting from l. 200. You talk about temperatures. How much the studies have taken account where the lichen comes from – high or low latitudes? The same about pH (and humidity also). Did the studies took account of original pH of lichen substrate (I mean calcareous substrate and bark of deciduous trees vs. acidic substrata and trees with bark of low pH value).

(A) In our review this is pointed out as a design deficiency even though it was not written so directly. Many authors do not take into account the physicochemical conditions of the natural environment for the selection of cultivation conditions, or at least they are not explicit in the reviewed works.

Culture media

(R1) It will be nice to compare graphically the isolation success depending on medium with different nutrition composition. i.e. from Corn Meal to Potato dextrose to …. Is there any correlation about the composition of the medium and the substratum where the lichen grows or what photobiont(s) it uses. I guess no.

It is not possible for us to compare the success of isolation and growth of different species that have been cultivated under different conditions and in many cases it is not openly stated in what sense this successful growth has been.

Conclusions

(R1) Conclusions are too short and for me does not carry necessary information. I would take all chapters and include point-by-point the most important message from the chapters.

(A) They have already been reviewed

(R1)  The most valuable part of the ms is the Table, however, I would do following modifications: 1) replace the column either that of order or family with the column of the class. 2) It would be easier to follow and it will be more useful if physico-chemical parameters are divided between different columns (temperature, light, pH, etc. ), also percentage of sugar can be in separate column.

(A) The class to which the species belongs was included in the table in the Class/Order column.

(R1)  The style of references’ list should be unified. Check the style in the instructions of the journal.

(A) They have already been reviewed

Regards

The authors

Reviewer 2 Report

I believe the title of the article does not accurately reflect the content of the manuscript. It does not critically analyse the methods and approaches, but merely states the data available in the literature. I suggest using the word 'review' instead of 'revision'.

It is my contention that this review fails to address two points of significant importance, if not outright criticality. The initial point to be addressed is the proof that the fungi obtained in culture are mycobionts. To provide such proof, it is sufficient to sequence the ribosomal genes and compare the resulting sequences with those from the Mycobank and NCBI (or alternative databases). Secondly, there is a lack of evidence to demonstrate that it is the mycobiont that forms lichen acids in some cases. In my opinion an additional section should be included before the "Conclusion" section, in which these two points should be subjected to critical analysis. It is worth noting that the author of this critical review has encountered the issue of contamination on multiple occasions when utilising the explant technique and ascospore isolation technique. In the publication by Cometto et al. (2024) (https://doi.org/10.1016/j.funeco.2024.101331), the focus is on lichen mycophilia and the significant diversity of endothallic and exothallic fungi in thalli. It is therefore imperative to confirm that a fungus isolated in pure culture is indeed a mycobiont. This issue merits further discussion in the context of this review.

The 'Conclusion' is somewhat lacking in detail and does not present the key findings on the relationship between the isolation technique and cultivation conditions and the success (or failure) of obtaining pure cultures of mycobionts. Furthermore, it does not include a conclusion on the confirmation of the affiliation of the obtained pure cultures to the corresponding species of lichenized fungi.

It is recommended that a column be added to Appendix Table 1A containing the mycobiont sequence accession number, where available.

Please be sure to format all names of genera, species, families, and other taxonomic ranks in the text of the manuscript and in Table 1A of the Appendix according to the recommendations (https://imafungus.biomedcentral.com/articles/10.1186/s43008-020-00048-6).

L278 – 279 It is important to note that stearic acid, linoleic acid, oleic acid, and other fatty acids are ubiquitous within cellular structures, including cell membranes. Consequently, any data pertaining to these acids should be subjected to rigorous scrutiny, particularly in light of the analytical techniques employed. The aforementioned acids can be extracted from all lichens using lipophilic solvents.

L11 'metabolites many' – please insert and or ; or another binding word

L12 'potential source' should by in plural

L27-29 'obligate biotrophs fungi (the mycobiont) that acquire fixed carbon' change for 'obligate biotrophic fungi (the mycobionts), which take up fixed carbon'

L58 after the word 'acetylcholinesterase' provide a reference to literature

L78-83 I recommend change for 'The development of in vitro culture methods for lichen-forming fungi would facilitate the large-scale synthesis of secondary metabolites, enhance our understanding of the role of partners in their synthesis, and increase our knowledge of genes connected to secondary metabolite production, thereby improving heterologous expression systems.'

L87-89 This conclusion is unsupported by a citation to a source and therefore appears to be illogical. For instance, Cladonia stellaris exhibits a relatively consistent microbial cenosis and mycobiont-edifier across different climate zones. The sentence is challenging to comprehend. An alternative formulation is provided below for consideration. ' The ability of lichen-forming fungi species of the same genus to flourish in a range of ecological environments represents a significant challenge for the standardization of methods and techniques for lichen culture at the genus level.'

L240 – Consider changing ‘Based on our review’ – ‘In accordance with the findings of the literature review’ or ‘the analysis of previously obtained data’.

L250-251 It is both logical and necessary to provide data on light intensity in accordance with the cited sources, as well as to indicate the effect that these two factors have on the growth of fungi.

L262 The sentence is devoid of a verb. One may either join it to the previous one or use a verb construction.

Author Response

August 12, 2024

Dear Reviewer 1,

We appreciate all your comments about the manuscript, which help us improve our work.

(R2) Major comments:

I believe the title of the article does not accurately reflect the content of the manuscript. It does not critically analyse the methods and approaches, but merely states the data available in the literature. I suggest using the word 'review' instead of 'revision'.

(A) The proposed change has been accepted

(R2)It is my contention that this review fails to address two points of significant importance, if not outright criticality. The initial point to be addressed is the proof that the fungi obtained in culture are mycobionts. To provide such proof, it is sufficient to sequence the ribosomal genes and compare the resulting sequences with those from the Mycobank and NCBI (or alternative databases). Secondly, there is a lack of evidence to demonstrate that it is the mycobiont that forms lichen acids in some cases. In my opinion an additional section should be included before the "Conclusion" section, in which these two points should be subjected to critical analysis. It is worth noting that the author of this critical review has encountered the issue of contamination on multiple occasions when utilising the explant technique and ascospore isolation technique. In the publication by Cometto et al. (2024) (https://doi.org/10.1016/j.funeco.2024.101331), the focus is on lichen mycophilia and the significant diversity of endothallic and exothallic fungi in thalli. It is therefore imperative to confirm that a fungus isolated in pure culture is indeed a mycobiont. This issue merits further discussion in the context of this review.

(A) The performance of genetic analyzes to confirm the mycobiont has been incorporated in chapter 4, as a fundamental part of the design of the experiments.

The 'Conclusion' is somewhat lacking in detail and does not present the key findings on the relationship between the isolation technique and cultivation conditions and the success (or failure) of obtaining pure cultures of mycobionts. Furthermore, it does not include a conclusion on the confirmation of the affiliation of the obtained pure cultures to the corresponding species of lichenized fungi.

(A) They have already been reviewed

(R2)It is recommended that a column be added to Appendix Table 1A containing the mycobiont sequence accession number, where available.

(A) Many authors do not mention it, which limit its inclusion within our review.

Please be sure to format all names of genera, species, families, and other taxonomic ranks in the text of the manuscript and in Table 1A of the Appendix according to the recommendations (https://imafungus.biomedcentral.com/articles/10.1186/s43008-020-00048-6).

(A) They have already been reviewed

L278 – 279 It is important to note that stearic acid, linoleic acid, oleic acid, and other fatty acids are ubiquitous within cellular structures, including cell membranes. Consequently, any data pertaining to these acids should be subjected to rigorous scrutiny, particularly in light of the analytical techniques employed. The aforementioned acids can be extracted from all lichens using lipophilic solvents.

(A) it is a wise suggestion that we appreciate you making

 (R2) Detail comments:

L11 'metabolites many' – please insert and or ; or another binding word

L12 'potential source' should by in plural

L27-29 'obligate biotrophs fungi (the mycobiont) that acquire fixed carbon' change for 'obligate biotrophic fungi (the mycobionts), which take up fixed carbon'

L58 after the word 'acetylcholinesterase' provide a reference to literature

L78-83 I recommend change for 'The development of in vitro culture methods for lichen-forming fungi would facilitate the large-scale synthesis of secondary metabolites, enhance our understanding of the role of partners in their synthesis, and increase our knowledge of genes connected to secondary metabolite production, thereby improving heterologous expression systems.'

L87-89 This conclusion is unsupported by a citation to a source and therefore appears to be illogical. For instance, Cladonia stellaris exhibits a relatively consistent microbial cenosis and mycobiont-edifier across different climate zones. The sentence is challenging to comprehend. An alternative formulation is provided below for consideration. ' The ability of lichen-forming fungi species of the same genus to flourish in a range of ecological environments represents a significant challenge for the standardization of methods and techniques for lichen culture at the genus level.'

L240 – Consider changing ‘Based on our review’ – ‘In accordance with the findings of the literature review’ or ‘the analysis of previously obtained data’.

L262 The sentence is devoid of a verb. One may either join it to the previous one or use a verb construction.

(A) All the suggestion above have already been accepted and incorporated

L250-251 It is both logical and necessary to provide data on light intensity in accordance with the cited sources, as well as to indicate the effect that these two factors have on the growth of fungi.

(A) This factor has not been extensively discussed by researchers since they have only included the factor but the design of the experiments they have carried out does not allow them to attribute differences in growth due to light intensity. In appropriate experimental design to study the influence of light intensity on crops, it is necessary.

Regards

The authors

Round 2

Reviewer 2 Report

I am grateful to the authors for substantial corrections to the manuscript text that have improved the quality of the manuscript. But...

My request to point out that there is no confirmation in any of the publications that a pure culture of mycobiont was obtained was effectively ignored by the review authors. I carefully again selectively analysed the publications (references 52, 53, 55, 60, 61, 66, 67, 68, 69, 92) and was convinced that none of them contains data on sequencing of obtained fungal cultures. Moreover, some publications carefully circumvent this issue by sequencing pure cultures of associated mycelial fungi or pure cultures of algae, and provide photos of colonies that do not belong to the mycobiont. I therefore insist that at the end of section 4 (Mycobiont isolation) it is necessary to describe this problem. The text on lines 209 - 213 should be placed in section 4, because it is the mycobionts that we are talking about. This issue cannot be approached formally, it is very important. Despite the abundance of publications on obtaining pure cultures, none of them gives strict evidence of axenicity and none of them contains sequences deposited in NCBI. If such data do exist and I have missed them, they should be provided. And if they do not, write that they do not exist. It is also important to mention that contaminating (associated) fungi, yeasts and bacteria are more likely to grow on the media used when growing from soredia or fragments of thalloms (rather than individual ascospores), even if the source material is thoroughly washed.

No comments.

Author Response

August 12, 2024

Dear Reviewer 2,

Once again we appreciate your comments, which have allowed us to substantially improve the manuscript.

(R2) Major comments:

I am grateful to the authors for substantial corrections to the manuscript text that have improved the quality of the manuscript. But...

My request to point out that there is no confirmation in any of the publications that a pure culture of mycobiont was obtained was effectively ignored by the review authors. I carefully again selectively analysed the publications (references 52, 53, 55, 60, 61, 66, 67, 68, 69, 92) and was convinced that none of them contains data on sequencing of obtained fungal cultures. Moreover, some publications carefully circumvent this issue by sequencing pure cultures of associated mycelial fungi or pure cultures of algae, and provide photos of colonies that do not belong to the mycobiont. I therefore insist that at the end of section 4 (Mycobiont isolation) it is necessary to describe this problem.

(A) The articles included in the present review were carefully reviewed and it was found that only 40% of the species had their identity confirmed by genetic testing. This data was recorded in Table A1, indicating the articles in which the molecular analyzes were carried out.

(R2) The text on lines 209 - 213 should be placed in section 4, because it is the mycobionts that we are talking about.

(A) Text was moved to section 4

(R2) This issue cannot be approached formally, it is very important. Despite the abundance of publications on obtaining pure cultures, none of them gives strict evidence of axenicity and none of them contains sequences deposited in NCBI. If such data do exist and I have missed them, they should be provided. And if they do not, write that they do not exist.

(A) Table A1 shows (through + symbol) the articles in which the molecular analyzes were carried out.

It is also important to mention that contaminating (associated) fungi, yeasts and bacteria are more likely to grow on the media used when growing from soredia or fragments of thalloms (rather than individual ascospores), even if the source material is thoroughly washed.

(A) That information was included in the text moved to section 4

Regards

The authors